# Comparing Contact Education and Digital Distant Pedagogy Strategies: Lockdown Lessons Learnt for University-Level Teacher Education

**Hannu Salmi \***, **Ninja Hienonen, Laura Nyman**, **Arja Kaasinen** and **Helena Thuneberg**

Department of Education, University of Helsinki, 00100 Helsinki, Finland
* Correspondence: hannu.salmi@helsinki.fi

**Abstract:** Teaching and learning experienced a rapid change in spring 2020, and the learning environments for university students changed almost overnight. An integrative science centre education informal learning course for Finnish teaching students has been arranged for over 20 years with latest technologies and innovations. This cross-sectional study compared survey data of teaching students between four time points: in 2019 ($n = 108$), in 2020 ($n = 115$), in 2021 ($n = 110$), and in 2022 ($n = 90$). The course content was the same, only the implementation differed. In the spring of 2019 and 2022 the course was implemented as contact-teaching, but during the critical phase of the pandemic in 2020 and 2021 as distant teaching. Data were analysed by using ANOVA and the prediction of possible effects of the contact/distance learning by the structural equation path model (SEM). The analysis showed that the results favoured the first contact instruction course in 2019: their confidence of integrating the learned contents of the science centre into practical school matters differed from all the other groups. This group also appreciated the usefulness of the course more than the other groups. In turn, the first distant course had a more negative opinion of the usability of the course than all the other groups. Despite that the distant group in 2020 and even more so the other distant group in 2021 felt more confidence in the direct integration of the course content into future teaching based on the path analysis. Gender had two kinds of effects, one in the distance learning group in 2020: being a female directly predicted the future use of science centre type pedagogy, and the other in the contact learning group in 2022: being a female predicted the appreciation of the science centre course. As a limitation of the study, more students' prior experience and attitudes with online learning is needed from future research. The preliminary results and best practises of this study are utilised internationally in several EU-Erasmus+ projects.

**Keywords:** e-learning; contact teaching; remote teaching; informal learning; teacher education

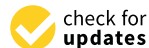



## 1. Introduction

Universities, along with schools, experienced a rapid change due to the pandemic crisis in spring 2020, and the learning environments for university students were changed almost overnight. The traditional way of organising teaching as contact instruction was replaced by the distance instruction via different online platforms, which proved to be a difficult task for instructors to plan their teaching to be suitable for online environments, and come up with new ways of assessing students' progress and achievements solely in online settings. For the purpose of this study, we refer to the instruction provided by educational institutions during the exceptional circumstances caused by COVID-19 in spring 2020 as Emergency Remote Teaching (ERT), as distinct from "usual" distance online instruction, as it has been provided primarily for the purpose of instructional support in an immediate manner during exceptional circumstances [1]. As it has been reported in several studies and literature reviews [2], in 2021 this course of action continued because of the long pandemic and the remote teaching turned out more as a new routine [3].

The effects on learning and teaching processes have differed depending on the different areas [4], school level [5], age groups, and gender even globally [6]. There have been some success stories related to digital distance learning, but also several failures in all educational levels, which also have not been reported [7].

In teacher education and professional development, one of the growing trends has been the use of open learning environments and informal learning sources [8,9]. The use of digital, web-based, remote, virtual, and virtual-augmented-mixed reality solutions is becoming a routine in this field. Furthermore, the exceptional times during 2020 showed certain potential for this type of Zoom-Teams-education. However, most of the studies related to informal learning have focused on basic education pupils. The main results show clear evidence of an increased motivation, interest, attitudes, and certain cognitive learning [10]. However, far less attention has been paid to teacher education and teacher–students' ability and knowledge on how to integrate informal learning in their future work in schools. The dilemma is that the teachers must know the topic they are teaching thoroughly before they can teach it to the students. This matters also for the educational practises; e.g., digital technologies and user friendly solutions. This is the main topic and approach of this study.

## 2. Contact Versus Distance Instruction

Due to these exceptional circumstances, the well-being and learning of students on all educational levels have increasingly been the focus of global media and research. Distance online instruction as such is not a new design and it is widely used, especially in higher education around the world, as it enables diverse ways to design and take courses independent of time and space [11,12]. Despite the better baseline in using online resources in higher education, polarised effects have been indicated: decreased motivation amongst students, challenges in self-directed learning, and increasing burnout rates, especially amongst younger students [13,14], who have been found to be more concerned about their mental well-being, economic situation as well as their career or studies compared to older people during COVID-19 [15]. In contrast, some students have been very motivated and engaged in their studies [16]. The concern of equality between different educational fields has also been prominent in Finland as some courses and lab work require contact learning, and because of this students' studies may be delayed. This can also have an influence on why there are different reactions to remote learning/new situations; some people may find it more fitting to their situation, and some find it difficult to learn independently without face-to-face instruction. However, as many instructors have implemented synchronous instruction (Zoom, etc.), people who usually find online learning more fitting may consider ERT as unsatisfactory because it is no longer possible to influence the timing of their studies and this might interrupt their life situation (children in home schooling, etc.) [17].

During these circumstances, students are forced to take the courses online and evidently students that have the choice to pick online courses are more determined and motivated to complete online courses, and the online courses may fit their lifestyle better [18]. However, it is necessary to consider the research concerning distance instruction in general, what kind of differences there are between face-to-face instruction, and thus see what might explain specific conditions during ERT.

All in all, there is a vast amount of research concerning online distance instruction and its relation to face-to-face instruction, especially concerning students' course satisfaction, [19–23], motivation [18,24–27], students' self-efficacy [18,22,23,28], learning outcomes [23,29], and their relation to interactional factors [19,21,30], learning styles [31,32], and self-regulated learning [22,23]. Research has indicated that student satisfaction in online learning is affected by a number of factors, such as students' self-efficacy beliefs [19,22], perceived e-learning usefulness [22], and the students–technology relationship (students' proficiency with the necessary technology and the usability of the technology) [21]. In addition, students that use online environments compared to traditional conditions (face-to-face instruction) tend to perceive themselves more able and are motivated and capable of

performing self-regulated learning [18,22,33]. Online learning has in fact been considered to require a high degree of autonomy, self-regulation, self-motivation, and self-direction [34]. It has also been discovered that students themselves appreciate the possibilities and support for the use of metacognitive self-regulation strategies and self-regulated learning that online environments offer compared to face-to-face learning sessions [13,33]. Prior experience seems to have a prominent role as well. According to Artino [23], the students' decision to take online courses depends on their self-efficacy beliefs about learning online and whether they have had satisfactory previous experiences with online courses. Students with prior experience in online learning have also been found to have more effective learning strategies, to be more motivated, have more positive self-efficacy beliefs about technology (and online learning systems) [28], and ultimately have higher grades compared to students without prior experience in taking online courses [23].

However, the way the online course is designed is a key factor in the way that students perform, are motivated, and satisfied with the online courses [26]. From the interactional point of view [35], how the learner–content interaction and learner–instructor interaction is designed affects the course satisfaction the most when compared to the interaction between peers [19,21]. The importance of the learner–content interaction may be due to the fact that online learners spend most of their time processing information by themselves [36], and therefore the materials of the course should be well designed in terms of stimulating interest, helping in the connection between new knowledge and the student's personal experiences, and the accessibility of the course content [19]. According to Chen and Jang [25], ill-designed online courses in terms of instructors' contextual support (for autonomy and competence) can even have a negative effect on students' outcomes. Wang et al. [26] have reported lower scores in the learning climate and psychological need satisfaction, and significantly higher scores in the need dissatisfaction among students in online courses compared to students in face-to-face courses. In line with previous research, their results revealed that the need satisfaction (feelings of autonomy, perceived competence, and relatedness) was related to knowledge transfer and thus higher-order thinking. In turn, need dissatisfaction was related to course grades, which would indicate vulnerability for more external motivation. Wang et al. [26] hypothesise that the physical and psychological distance (transactional distance) may be one of the reasons for these differences. In addition, even if the asynchronous features of online learning can be beneficial due to its flexibility, it can also be a challenge for providing social support which is crucial from the viewpoint of students' motivation, learning outcomes, and well-being. As Zhan and Mei [37] have found, the greater effect of the social presence on academic achievement in online courses could be due to the fact that social presence can be more built-in, have informal forms, be direct and free, and go beyond words to body language more in the face-to-face instruction. Therefore, online learners can also feel more isolated [38,39] and disengaged [40].

Paechter and Maier [12] have stated that different learning objectives and learning processes can be better supported in online and face-to-face instruction, which indicates that there might be characteristics in every instructional category that are better suited for one than the other. Based on their results, when fast exchanges and communication are needed between the students and instructor or peer learners, students prefer online contact. In turn, when the subject matter demanded conceptual knowledge building with the exchange of ideas and the application of the conceptual knowledge in practise, students preferred face-to-face contact. In particular, an interaction and a discourse in which a shared meaning is constructed (which was considered to be better offered in face-to-face instruction) were emphasised as crucial for the course satisfaction overall.

The possible performance gaps between online and face-to-face courses have also been thought to be related to academic subject areas, as Xu and Jaggars [28] found that within social sciences (e.g., psychology) amongst the applied profession (business, law, and nursing), the gap was greater as these areas may consist of more hands-on demonstrations and practise, and thus can make effective online materials, activities, and assignments harder to create for the instructor.



It is noteworthy that online learning and face-to-face learning and their effects on the engagement and achievement can be different. It seems that the focus needs to be more on the question of whether the students learn what was intended and not on how well students learn compared to traditional methods, since despite the different benefits of online learning, it has also raised concerns regarding low student engagement and high dropout rates in online courses [41]. In addition, the whole eco-structure of course design should be considered similar to the resources of libraries and other information sources. Ultimately, it all comes down more to the pedagogical skills of the instructors to use online recourses that are suitable for reaching the goals intended than to the fact that the learning and instruction is not happening in the same physical space.

## 3. Informal Learning

Informal learning has its origins in the proponent of radical pedagogy such as Rousseau [42], with its modern roots in the works such as *Deschooling Society* by Ivan Illich [43] or *The Unschooled Mind* by Howard Gardner [44]. The terminology was clearly defined for the first time in the UNESCO report *Learning to be* [45]. This terminology concerning the division into formal education and informal learning has been accepted in the literature for decades [46]. In the 1990s, science centres were strongly developing interactive, hands-on pedagogy [47]. In the 2000s, these ideas have become widely accepted [48] and integrated into school education, especially regarding science education [10]. Out-of-school education is a term meaning teaching according to the official curriculum but utilising informal learning sources outside the physical school settings [8]. In addition, the amount and quality of educational, evidence-based research has been growing [10]. Bridging the gap between formal education and informal learning has shown its potential during the digital era [49]. Originally, online solutions for learning were developed mainly in informal learning settings [50]. In addition, the very first formal education contents utilised materials created for other-than-direct educational purposes [51]. The rapid change during the past year has given an impetus to develop ways to arrange informal learning as a means of remote teaching.

One natural place to provide informal learning is a science centre. A science centre is a learning laboratory in two senses: First, it is a place where a visitor can learn scientific ideas by themselves using interactive exhibition units. Secondly, it is a place where informal education can be studied in an open learning environment [52]. In this study, the science centre forms the platform for testing and evaluating the new National Curriculum of Finland with the new tasks of seven transversal competences.

## 4. Transversal Competence Areas

Crises in societies—such as the Sputnik phenomenon in the 1960s, the Chernobyl accident in 1985, Global Climate Change, and the COVID-19 pandemic since 2020—have also had a huge impact on schools and educational systems [34,47,53,54]. Similar to the informal learning initiative initiated by UNESCO in the 1960s and the "life-long learning" terminology from the working life needs of the 1980s, the movement for so-called "21st Century Skills" also started globally on the initiative of the OECD and several active foundations with relationships to business life. The question and challenge that remains is, if and how the educational system can renew itself and respond to these global challenges.

In Finland, seven transversal competence areas were included in the National Core Curriculum in 2016 [55]. The Finnish version of the internationally shared idea of 21st century competences and the need for these competences arises from changes in the surrounding world. The areas are seen as principles that guide all subject-specific instruction and must also be addressed through multidisciplinary learning modules [50]. The seven areas of transversal competences (T1–T7) are said to form an entity of knowledge, skills, values, attitudes, and will. Competences become realised as abilities to apply knowledge and skills appropriately when needed. The seven competence areas are: *Thinking and learning to learn (T1)*, *Cultural competence, interaction and self-expression (T2)*, *Taking care of*

oneself and managing daily life (T3), Multiliteracy (T4), Information and communication technology (ICT) Competence (T5), Working life competence and entrepreneurship (T6), and Participation, involvement and building a sustainable future (T7). In addition to every school subject, one context by which to integrate the competences is through multidisciplinary learning modules that allow pupils to see interdependences between phenomena and to build meaningful knowledge entities in interactions with others. Informal learning sources and out-of-school settings seem not only to offer fruitful contents of teaching transversal competences but are also a necessary context for this type of future competences [55].

## 5. The Multidisciplinary Course

Every year since 1994, more than 5000 teaching students from the University of Helsinki have participated in the Integrative science education course (5 ECTS points), as a part of their compulsory academic studies. The core of the course is one whole day of studies in the science centre with hands-on experiments, interactive exhibits, a modern science exhibition, workshops, laboratories, demonstrations, and a planetarium. The course consists of pre- and post-lectures by pedagogical experts and professors. The aim of this multidisciplinary science learning course is to bridge the gap between formal education and informal learning via teachers' education. Furthermore, the aim of the course is to develop ways to teach transversal competences by means of informal learning.

*Didactics of the Multidisciplinary Learning Module Course*

The National Core Curriculum in Finland indicates that there has to be at least one multidisciplinary learning module (MLM) every school year [55]. That means a clearly defined theme, project, or course where different subjects are combined and the selected theme is approached from the perspective of several subjects. The duration of these modules can vary from a day to weeks [56].The National Core Curriculum for basic education is also a basis for the primary school teacher education curriculum at universities. Both curricula are undergoing reforms of different magnitudes every few years to react to the needs of competences for the twenty-first century [57]. The topics to be taught must be aligned with the national educational guidelines in teacher education. Furthermore, each school's subjects should be studied usually from a didactical or pedagogical point of view. The didactics of the multidisciplinary learning module (MLM) is taught at the end of the second studying year at the University of Helsinki. Before this course, teaching students have studied all other school subject courses such as the didactics of biology, mathematics, arts, and such and they have the opportunity to create an MLM by themselves. The idea of this is to increase and support teaching students' professional skills for teaching.

The findings and the results of the earlier courses can be summarised as follows:

1.  Moving from teacher-controlled learning to pupil-oriented learning with context-related knowledge shows clear changes in the roles and responsibilities of both the teachers and pupils [58].
2.  Findings related to informal learning and its relevance indicate that the crucial period for early professional development seems to be the first three years as an inexperienced teacher [59].
3.  Having pre- and post-materials, information, and lectures are essential to reach permanent and cost-effective pedagogical solutions [60].
4.  Using programs linking the school curriculum and science centre exhibitions, encouraging results were achieved among the teachers and teaching students [61].
5.  With digital, virtual, and augmented reality (AR) solutions [62] it is possible to combine real objects with virtual ones and to place suitable information into the real surroundings. It is also noteworthy that the teachers were not impressed with the technology itself but with seeing ICT as a connection to the learning environment, an instructional tool. This can lead in the best case—according to the teachers' interviews—to changes in the roles and responsibilities of pupils and teachers [63].

So far, both parts of the course, the lectures, and a one-day visit to the science center, have been organised on-location. In the spring term of 2020, the course started as a normal on-location course. However, midway through the course, the implementation needed to be changed as the university recommended transitioning all teaching online. In contrast to the course in spring 2019, the visit to the science centre and the lectures before and after the visit were implemented online. Instead of the interactive hands-on exhibition, the students received replacement tasks on the same STEAM-topics related to science, astronomy, chemistry, the environment, engineering, technology, biology, architecture and art skills, history, and math via Zoom during the eight-hour day with short breaks. The pedagogy of these tasks was based on subject-orientated integration and the seven transversal competence areas were included, as presented earlier.

## 6. Aim of the Present Study and Research Methodology

Bridging the gap between the formal education and informal learning via teacher education that is online an on-location is the main approach of this study. This is done by comparing teaching students' survey data from the same course that was implemented in four time points: 2019 on-location, in 2020 and 2021 online, and 2022 on-location.

The research questions are:

1.    Is there a difference between the contact and distance instruction groups in their understanding of the new concept of transversal competence areas?
2.    Is there a difference between the contact and distance instruction groups in their confidence in applying the type of pedagogics learned to teaching in the future?
3.    How useful would the students rate the science centre learning module and does this vary by the type of instruction, contact and distance?
4.    Of interest: will the possible novelty effect of distance learning hold after a period, when it is not anymore so new?

### 6.1. Research Participants

The data consisted of 423 teaching students, divided in four groups: the contact instruction group in 2019 ($n = 108$), the distance instruction group in 2020 ($n = 115$) and 2021 ($n = 110$), and the contact instruction group in 2022 ($n = 90$). In total, there were 359 females (85%) and 63 males (15%) (missing of 1 case). The majority of the students were 25 years old or younger (59%) and the rest were 25 to 30-year-olds (19%) and those over 30 years (22%). Of the students, 31% had no working experience as a teacher, 43% less than a year, 19% one to five years, and 8% had been teaching for more than five years.

### 6.2. The Survey Instrument

The survey was administered during the course as part of a normal course assignment. It was distributed through an online survey platform. The data on the contact instruction group were collected in April 2019, the distance instruction groups in April 2020 and 2021, and the contact instruction group in April 2022. All surveys included the same questions. The survey included the background questions presented above and self-reported statements on science teaching and transversal competence areas. The scale for statements concerning the easiness of transversal competence areas to be integrated in science teaching were: 1 = very easy; 2 = fairly easy; 3 = not easy or difficult; 4 = fairly difficult; 5 = very difficult, and for the statements concerning the usefulness of the course: 1 = very useful; 2 = fairly useful; 3 = I cannot decide; 4 = fairly useless; 5 = very useless. For the analyses, the scales were reversed.

### 6.3. Statistical Analysis Methods

Firstly, the descriptive statistical data were analysed and the differences between the contact and distance instruction groups were compared using one-way analysis of variance (ANOVA) in concepts related to the transversal competences and in the confidence in integration of the pedagogics in practise. For that, two composite variables were

formed: (1) transversal competence integration (formed from the T1–T7 competences, see Table 1), α = 0.751, 7 items; (2) integration in practise (formed from the variables *How easy to integrate the multidisciplinary science teaching module contents with curriculum/school timetable/preparation time*) α = 0.610, three items. Secondly, to answer the question more thoroughly and explore the relative role of the variables, a structural equation path model (SEM) was created. The Chi-square value, Tucker–Lewis Index (TLI), Comparative Fit Index (CFI), and Root Mean Square Error of Approximation (RMSEA) were used to assess the goodness-of-fit. SPSS Statistics 27 and AMOS 26 were used for the analyses.

**Table 1.** The descriptive statistics of the Likert-scale (1–5) variables.

| | *N = 423* | | *Contact 2019 n = 107* | | *Distant 2020 n = 115* | | *Distant 2021 n = 110* | | *Contact 2022 n = 90* | |
|---|---|---|---|---|---|---|---|---|---|---|
| | *M* | *SD* | *M* | *SD* | *M* | *SD* | *M* | *SD* | *M* | *SD* |
| How easy will it be to integrate the multidisciplinary science teaching module contents with *the curriculum* | 3.93 *** | 0.70 | 4.34 | 0.61 | 3.94 | 0.64 | 3.73 | 0.59 | 3.66 | 0.75 |
| … with *school timetable* | 3.48 * | 0.74 | 3.67 | 0.75 | 3.40 | 0.72 | 3.46 | 0.70 | 3.37 | 0.79 |
| … with *preparation time* | 3.27 *** | 0.84 | 3.42 | 0.86 | 3.46 | 0.78 | 3.16 | 0.85 | 2.98 | 0.81 |
| How likely are you to use the contents of the multidisciplinary science teaching module in teaching in the next 5 years | 3.44 | 0.75 | 3.50 | 0.74 | 3.37 | 0.71 | 3.47 | 0.76 | 3.42 | 0.81 |
| How easy will it be to integrate the multidisciplinary science teaching module contents with *Thinking and learning to learn (T1)* | 4.25 | 0.69 | 4.28 | 0.70 | 4.37 | 0.60 | 4.20 | 0.70 | 4.14 | 0.77 |
| … with *Cultural competence, interaction and self-expression (T2)* | 3.86 *** | 0.77 | 3.69 | 0.82 | 3.70 | 0.82 | 4.04 | 0.65 | 4.06 | 0.68 |
| … with *Taking care of oneself and managing daily life (T3)* | 3.93 ** | 0.80 | 3.88 | 0.83 | 3.75 | 0.84 | 4.03 | 0.71 | 4.09 | 0.76 |
| … with *Multiliteracy (T4)* | 3.89 | 0.83 | 3.89 | 0.86 | 3.88 | 0.83 | 3.89 | 0.75 | 3.89 | 0.92 |
| … with *ICT Competence (T5* | 4.04 * | 0.80 | 4.08 | 0.81 | 4.21 | 0.72 | 3.92 | 0.84 | 3.93 | 0.82 |
| … with *Working life competence and entrepreneurship (T6)* | 3.34 ** | 0.95 | 3.10 | 0.92 | 3.26 | 0.92 | 3.51 | 0.93 | 3.50 | 0.99 |
| … with *Participation, involvement and building a sustainable future (T7)* | 4.11 ** | 0.80 | 4.30 | 0.73 | 4.19 | 0.76 | 3.94 | 0.84 | 4.00 | 0.84 |
| How usefulness of the time spent in the multidisciplinary science teaching module (lectures & visit) | 3.95 *** | 0.84 | 4.29 | 0.81 | 3.61 | 0.78 | 3.99 | 0.82 | 3.93 | 0.79 |
| *Transversal integration* (sum variable of 7 competences) | 3.92 | 0.51 | 3.89 | 0.52 | 3.91 | 0.48 | 3.93 | 0.49 | 3.94 | 0.58 |
| *Integration in practise* (sum variable of curriculum integration, timetable, preparation) | 3.56 *** | 0.57 | 3.81 | 0.56 | 3.60 | 0.54 | 3.45 | 0.50 | 3.33 | 0.60 |

Note: *n* = number of cases in subgroups, M = mean, SD = Standard deviation. * *p* < 0.05, ** *p* < 0.01, *** *p* < 0.001.

## 7. Findings with Discussion

*Statistical Descriptives and Group Differences*

Overall, the absolute means (Table 1) were high on the scale of 1 = very easy–5 = very difficult. When the seven transversal competence areas (T1–T7) were compared, the most confident students were with "integrating the multidisciplinary science teaching module contents with *Thinking and learning to learn (T1)*". The most deviation was found in the question "How easy will it be to integrate the multidisciplinary science teaching module contents with *Working life competence and entrepreneurship (T6)*".

The largest differences between the means of T1–T7, tested by ANOVA, related in the integration in practise (and its constitutes: curriculum integration, timetable, preparation) were found in Cultural competence, interaction and self-expression (T2). For the differences between the four groups in how useful they experienced the multidisciplinary science

teaching module and the time spent in it, the Contact 2019 group experienced the course as significantly more useful than the Distant 2020 group ($p < 0.001$) and the Contact 2022 group ($p < 0.05$) (also almost than the Distant 2021, $p = 0.058$). See Table 1 for the descriptives and the overall significant differences between the groups, Table 2 for the significant differences from the pairwise comparisons, and Figure 1 for the 95% confidence interval plots illuminating the bigger picture of variables in the four time points by the subgroups.

The comparison between the four subgroups from the different time points showed that the results were higher in the Contact 2019 course (see especially the 95% confidence interval plots in Figure 1) in relation to the *confidence of integrating the learned contents of the science centre module* into *practical school matters* (curriculum, time-table, and preparation time) in general. The difference was significant between it and the distance courses of 2020, 2021, and the later contact course in 2022. The students of the first contact course in 2019 also appreciated the usefulness of the course more than the later groups; the first distance course differed from all other groups with a more negative opinion.

**Table 2.** The significant mean differences between the subgroups.

| Compared Variable | Group vs. | Group |
|---|---|---|
| How easy will it be to integrate the multidisciplinary science teaching module contents with *the curriculum* | Conta19 > | Dista20 *** |
| | | Dista21 *** |
| | | Conta22 *** |
| | Dista20 > | Conta22 * |
| ... with *school timetable* | Conta19 > | Conta22 * |
| ... with *preparation time* | Conta19 > | Conta22 ** |
| | Dista20 > | Conta22 ** |
| How easy will it be to integrate the multidisciplinary science teaching module contents with *Cultural competence, interaction and self expression (T2)* | Conta19 < | Dista21 * |
| | | Conta22 * |
| | Dista20 < | Dista21 ** |
| | | Conta22 ** |
| ... with *Taking care of oneself and managing daily life (T3)* | Dista20 < | Conta22 * |
| ... with *Working life competence and entrepreneurship (T6)* | Conta19 < | Dista21 * |
| | | Conta22 * |
| ... with *Participation, involvement and building a sustainable future (T7)* | Conta19 > | Dista21 * |
| How usefulness of the time spent in the multidisciplinary science teaching module (lectures & visit) | Conta19 > | Dista20 *** |
| | | Dista21, $p = 0.058$ |
| | | Conta22 * |
| | Dista20 < | Dista21 ** |
| | | Conta22 * |
| *Integration in practise* (sum variable of curriculum integration, timetable, preparation) | Conta19 > | Dista20 * |
| | | Dista21 *** |
| | | Conta22 *** |
| | Dista20 > | Conta22 ** |

Note: Subgroups: Conta19 = Contact 2019, Dista20 = Distant 2020, Dista21 = Distant 2021, Conta22 = Contact 2022. The significantly bigger mean is indicated by sign >. * $p < 0.05$, ** $p < 0.01$, *** $p < 0.001$.

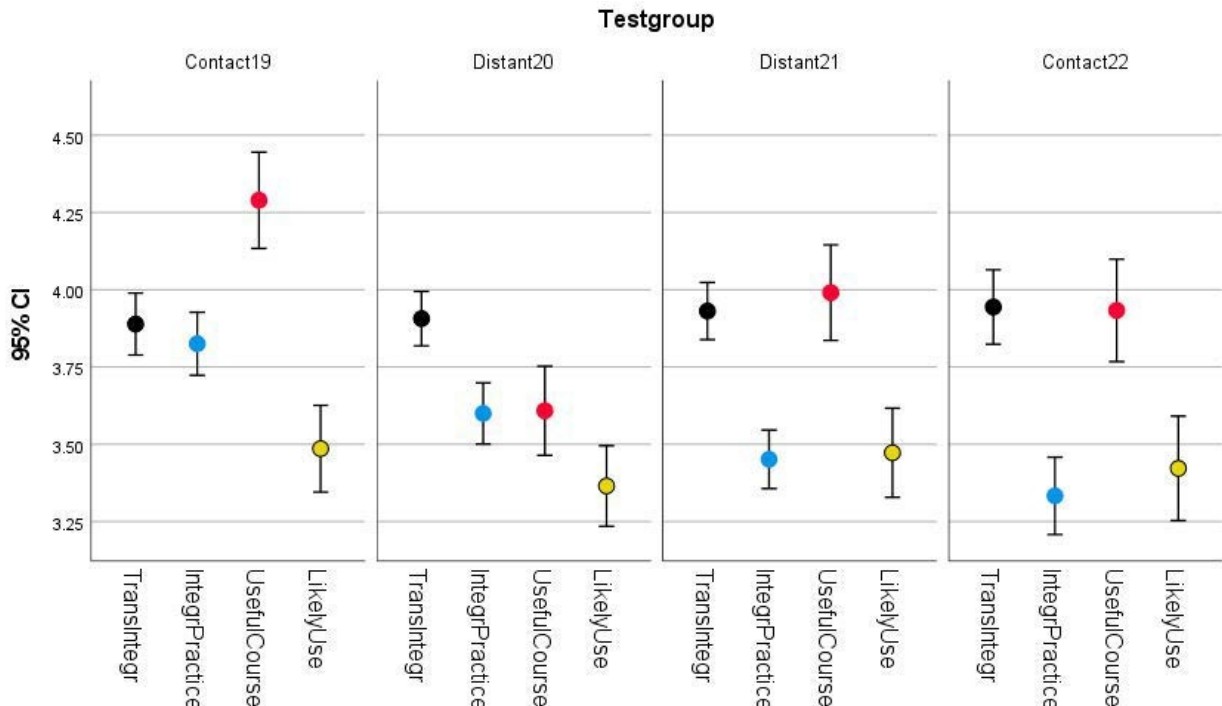

**Figure 1.** The 95% confidence interval plots of the composite variables Transversal integration and Integration in practise, and the variables Usefulness of Course and Likely to use in the future.

The Contact 2019 group was more confident in integrating the T7 (*Participation, involvement and building a sustainable future*) into teaching than the Distant 2021 group (Table 2). However, the two later groups, Distant 2021 and Contact 2022 groups, were more positive in relation to other competences, more sure of being able to integrate the T2 competence (*Cultural competence, interaction and self expression*) into teaching than the Contact 2019 and Distant 2020 groups. For the Contact 2019 group this was true also in relation to T6 (*Working life competence and entrepreneurship*), and the Distant 2020 group was less confident with the T3 (*Taking care of oneself and managing daily life*) than the Contact 2022 group.

In order to explore the relative statistical role of the variables and visualise the effects, the structural equation path model was created by using gender (1 = male, 2 = female), age (1 = less than 25 years, 2 = 25–30 years, 3 = more than 30 years), and *teacher experience* (0 = none, 1 = some, but less than one year, 2 = 15 years, 3 = over 5 years), as the exogenous (independent) factors. The endogenous (dependent) variables were *usefulness of time spent in the science centre course* and *how likely and often one will use science centre type pedagogy in the next 5 years*, and the composite variables were: *science centre course integration with transversal competences* and *integration of the science centre course in practise*.

## 8. Overall Results of the Path Modelling
### 8.1. Overall Results of the Path Modelling

As we earlier hypothesised, the same model would likely not fit with all the subgroups and accordingly an invariance test was conducted. The test confirmed that the subgroup was a significant moderator and the groups were different at the model level. The difference between the unconstrained and constrained model was: Chi-square (df = 36), *p* < 0.01. The analysis continued by testing the differences between the paths. The criterion of including the predictor in the model was that at least in one group the path was significant. The path models are displayed by the subgroup in Figure 2. Significant path arrows and the standardised betas (*β*) are marked by the red colour. The $R^2$ indicates the total standardised explanation of the predictors on the concerned variable. See also Table 2 for the statistical indicator z for the path differences and the effect size.

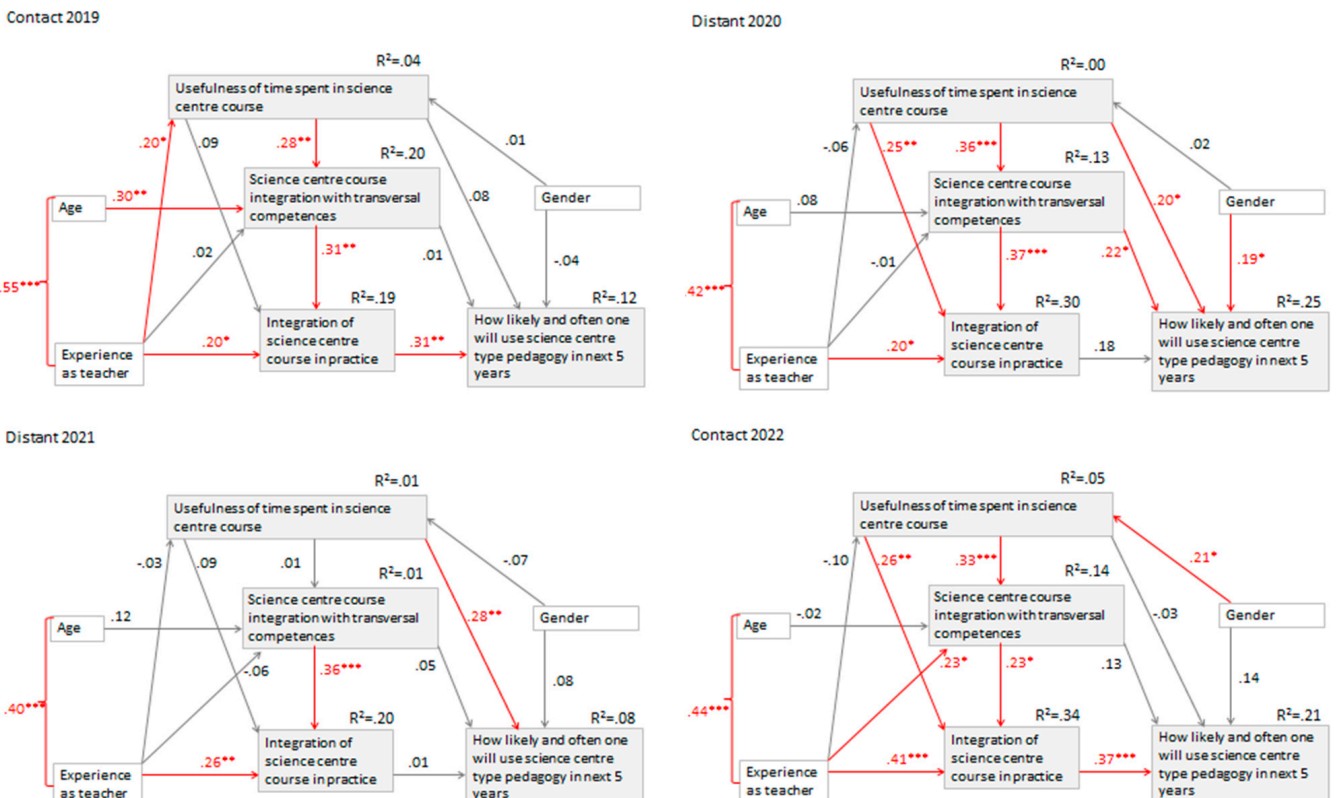

**Figure 2.** The path models of the four subgroups. (The significance of the predictors: * *p* < 0.05, ** *p* < 0.01, *** *p* < 0.001). See also Table 2 for the statistical indicator z for the path differences and the effect size.

The final model fitted the data well based on the Chi-square test and the goodness-of-fit indexes ($\chi^2$ = 30.351, *df* = 32, *p* = 0.550; *TLI* = 1.020; *CFI* = 1.000; *RMSEA* = 0.000).

The total standardised explanation of the variance on *Usefulness of time spent in science centre course* varied between the subgroup models ($R^2$ = from zero to 0.05). the significant predictors were (only in two models) *Experience as a teacher* and *Gender*. On *Science centre course integration with transversal competences*, the direct and indirect predictors were totally explained depending on the model: $R^2$ = from 0.01 to 0.20. The total explanation on *Integration of science centre course in practise* varied: $R^2$ = from 0.19 to 0.34. The explanation on *How likely and often one will use science centre type pedagogy in next 5 years* varied as well: $R^2$ = from 0.08 to 0.25.

The teaching experience had an effect on *Integration of science centre course in practise*, with the effect varying from $\beta$ = 0.20 * − 0.41 ***. The age had an effect in the contact group 2019 model: the older the teacher was, the more confidence in *integrating the learned course contents into transversal competences* they had *($\beta$ = 0.30 **)*. The gender had two effects, one in distant learning group 2020, in which being a female directly predicted the future use of the science centre type pedagogy ($\beta$ = 0.19 *), and the other in the contact learning group 2022, in which being a female predicted the *usefulness of the science centre course* ($\beta$ = 0.21 *).

*8.2. Significant Differences in the Paths between the Four Subgroup Models*

8.2.1. Integration of Science Centre Course with Transversal Competences

The *usefulness of time spent in science the course* path was significantly more powerful in the models of the Contact 2019 group ($\beta$ = 0.28, z = 2.189 *), the Distant 2020 group ($\beta$ = 0.36, z = 2.785 **), and the Contact 2022 group ($\beta$ = 0.33, z = 2.558*) than in the Distant 2021 group ($\beta$ = n.s.). *Experience as teacher* in the Contact 2022 group predicted it more effectively ($\beta$ = 0.23, z = 2.058 *) than in the Distant 2021 group in which there was not a connection at all ($\beta$ = n.s.). *Age* was a significant predictor in the Contact 2019 group

($\beta = 0.30$, z = 1.969) on the variable of concern, but had no role in the Contact 2022 group ($\beta$ = n.s.).

### 8.2.2. Integration of Science Centre Course in Practise

*Experience as a teacher* predicted the variable significantly more ($\beta = 0.41$) in Contact 2022 than in Contact 2019 ($\beta = 0.20$, z = 2.095 *) and in Distant 2020 ($\beta = 0.20$, z = 1.985 *).

### 8.2.3. How Likely and Often One Will Use Science Centre Type Pedagogy in the Next 5 Years

*Integration of science course in practise* predicted the variable more in Contact 2019 ($\beta = 0.31$, z = 2.14 *) and Contact 2022 ($\beta = 0.37$, z = 2.565 *) than in Distant 2021 ($\beta$ = n.s.). *Usefulness of time spent in science centre course* predicted the variable more in the Distant 2021 group ($\beta = 0.20$, z = 2.124 *) than in Contact 2022 ($\beta$ = n.s.).

The aim of this study was to compare the teaching students' confidence on learning new concepts and tools relating to 21st century competences being introduced in the curriculum and concretized in the science centre learning module. There were four groups, which were exposed to two different types of instruction, of which the first and last courses of the four experienced the more familiar contact type teaching, and the second and third courses were given a novel type, distance instruction. The latter likely or surely demanded a somewhat different attitude, motivation, and also skills than the traditional teaching method. In addition, psychologically the time of the pandemic was challenging, many self-evident everyday things ranging from meeting friends to being able to enjoy hobbies were restricted, which could have caused anxiety, stress, and loneliness. Those factors were not analysed in this study, but they might be a part of the results on understanding the concept of transversal competences and on the confidence in applying the type of pedagogics learned to teaching in the future using the type of instruction practised during the course.

The analysis showed that the results favoured the first contact instruction course in 2019. This became clearly evident in their confidence of integrating the learned contents of the science centre into practical school matters (curriculum, time-table, and preparation time) in general, in that the Contact 2019 group differed from all the other groups, including both distance courses of 2020, 2021, and the contact course of 2022.

Our aim, furthermore, was to compare how useful the students rated the science centre course. The results indicated that the students of the first contact course in 2019 appreciated the usefulness of the course more than the other groups. It is of interest that the students of the first distance course had a more negative opinion on the usability of the course than all the other groups. Despite this, the distant group in 2020 and even more so the other distant group in 2021, felt more confidence in the direct integration of the course content into future teaching. This was shown by the path analysis, which resulted in four models due to the significant moderation test.

When the background variables were taken into account, the teaching experience was related to the practical integration of the course contents in all models. The older the student teachers were, the more confidence they had on integrating the learned course contents into the transversal competences (only contact 2019 group). Gender had two kinds of effects, one in the distant learning group of 2020: being a female directly predicted the future use of science centre type pedagogy, and the other in the contact learning group of 2022: being a female predicted the appreciation of the science centre course. Furthermore, the age and teaching experience had a bigger role in the contact instruction groups than in the distant ones for that same variable.

Because the models were more or less unique and the paths were varying between the contact and distant learning groups, the explanation rates were also not the same:

- The smallest explanation of the variance was on *Usefulness of time spent in science centre course*, for which the maximum was 5%.
- On *Science centre course integration with transversal competences*, the direct and indirect predictors totally explained 1–20% depending on the model.
- The total explanation varied in the highest level on *Integration of science centre course in practise* from 19 to 34%.
- The explanation of the variance on How likely and often one will use science centre type pedagogy in next 5 years varied between 8 and 25%.

For some reason the perceived usefulness of the science centre course did not touch the confidence of *integrating concept of transversal competences into teaching* in the distant instruction group of 2021, but, on the contrary, in the distant group of 2020 the appreciation of the course did have a medium-sized effect.

On *the integration of science centre course in practise* the teaching experience showed the most powerful (yet a medium-sized effect) connection in the contact instruction group of 2022. In contact groups 2019 and 2022, the confidence in the ability to integrate the science centre course in practise predicted the future use of the science type pedagogy more than in the distant learning groups, significantly so in the distant group 2021. When the science centre module comprises concrete demonstrations and hands-on methods in situ, it seems natural that the contact groups were directed towards the future with practical tools.

In turn, the distant instruction groups, despite having the confidence in practical integration, did not realise its relevance and connection with future work. It is interesting that the effect on future use of the science centre learning contents went in the distant instruction groups either directly from experienced usefulness of the course or via the more conceptual variable *science centre course integration with transversal competences*, not via *the integration of science centre course in practise* like in the contact instruction groups.

### 9. Limitations of the Study

Some limitations concerning the present study need to be mentioned. There may be some differences between the four groups that were not investigated in this study, as the students were not randomly assigned to one or the other form of the course. Furthermore, the survey did not contain any questions about how much time students spent on the course. Previous research has shown that there is considerable variation between contact and distance teaching, and especially within the distance form of teaching in terms of how much time students spent on course work [64]. This, in turn, may explain the differences in experienced usefulness or confidence. In addition, more information on how the students experienced the teaching in the contact course and in the distance course is needed [20], as well as the information on students' prior experiences with online learning [21]. Furthermore, it would be useful to measure the students' attitudes towards technology use. For example, Dillon and Fraser [65] applied the Technology Acceptance Model (TAM), originally developed by Moore and Benbasat [66]. The TAM is one of the most well-known models in information systems research, developed to analyse and predict how users adopt information technology. It was designed to "measure users' perceptions of adopting an information technology innovation". In the future, this should also be taken into account when studying differences between the contact and distance instruction.

The type of instruction in the spring of 2020 course was changed unexcepted overnight, and there were no existing plans on how to implement the course and, for example, how to replace the science centre visit online. This also means that the findings of the distance2020 group are not fully comparable with previous distance instruction research.

### 10. Conclusions

What makes the findings of this study interesting is that the main results are not unified but also even conflict with each other. The three year-long period contained several changes due to the COVID pandemic and also changes in the rapid steps forward towards using different types of combinations of online and on-location teaching and learning.

Many capacities of the students were regressing, some were recovering, a few skills were even developing better. Normally, the result of the skills staying more or less the same is a "zero result", but in this case the fact that certain multidisciplinary capacities did not change during the long and critical COVID period was really a fruitful finding at least for future studies.

Usually, it requires time to plan and convert a contact-teaching course into an online course. However, in the spring term of 2020 both the lecturers and students needed to adapt to the new situation overnight. The results somewhat favoured the contact group, but the results were not straightforward. Students in the contact group experienced the course as being more useful. This may partly be explained by the visit to the science centre. It is possible that students doing the online task, on the other hand, did not experience the same interactivity as the students in the contact group. However, students in the distance learning group were more confident about integrating the transversal competences into their future teaching.

As a summary, from the total of fourteen items indicating teaching capacities before and after the COVID period of 2019–2022:

-    seven items showed clearly diminishing skills,
-    four items indicated the skills remained on the same level,
-    four items showed clearly better teaching skills.

Thus, many of the learning results from pre-COVID 2019 did not recover to the same level in the post-COVID time of 2022. There are several possible explanations for this declining and regressing process: one obvious reason might be that during the two pandemic years 2020–2021, the teaching students did not have the opportunity to learn certain skills and knowledge that were needed to utilise the content of this special course demanding certain basic capacities. For example, the students felt it was more difficult "to integrate the multidisciplinary science teaching module contents with the curriculum" all the time. The declining trend from 2019 to 2022 regressing during the process in a statistically significant manner was reported in the results earlier. It seems evident that the teaching students did not receive in their distance learning studies enough curriculum analysis skills to apply them in a practical situation. This finding calls for more research.

It is also most interesting to see that certain teaching capacities improved during and after the COVID period distance pedagogy. When the seven transversal competence areas and their integration into the science teaching module were studied, the results were slightly varying. Overall, this is in line with the previous research showing that the experiences from online and face-to-face courses may vary between different subject areas [28].

The analyses showed that the integration of the cultural competence, interaction and self-expression (T2), taking care and managing daily life (T3), and working life and entrepreneurship (T6) were improved during the later part of 2021–2022 of the COVID period. One reason might be that the rapid change to distance learning and the unexpected changes in the world gave the teaching students experience and skills helping them to learn certain 21st century skills and to realise their importance.

T2 especially had the clear improving tendency. It certainly has the roots in informal learning: Everyday experiences during the COVID time gave and demanded this type of competences and skills from everyone.

*Thinking and learning to learn (T1)*, *Multiliteracy (T4) and Participation, involvement and building a sustainable future (T7)* stayed practically at the same level during the whole time span. Especially the fact that the students did not feel their capacities related to *multiliteracy* improved was somewhat surprising: during the COVID period it would have been expected that the student would have learnt several skills related to different types of literacy, visual, and other types of materials. A somewhat unexpected result was also found with *ICT Competence (T5)* as the results were declining. It would have been expected that as the students were forced to utilise different ICT skills in their own studies, they would have also experienced it to be easier to integrate the T5 in the future teaching.

The experience of the students' capacity to integrate *T1 in future teaching* was practically in the same level. The explanation is most probably the "glass roof effect": the starting level was rather high—it could not develop much better—and the multidisciplinary course gave a firm basis to conserve these essential capacities. As there are no previous findings on this, the conclusions of this part of the results are merely speculations. However, they certainly provide opportunities for further evaluation and research.

Distance learning and pedagogy certainly have several advantages, especially cost-effectiveness in many senses. However, even some research-based evidence, especially related to the latest, brand-new technologies also tend to give false promises or create only ambitious future visions. However, the reality hits back: maybe it is better to speak about "digi-naives" instead of "digi-natives" [67]. This is more than true when we analyse the results related to ordinary Zoom/Skype/Teams types of distance learning. In addition, most of the results praising the positive and effective results related to distance learning during the exceptional times do not have comparative results or even quasi-experiment study designs.

In addition, both the teachers and the students have weaknesses in their practical routine skills and equipment. It also seems that the lecturers try to follow old-fashioned traditions while giving their lectures. In addition, the novelty in the very first phase in 2020 of the COVID period distance learning might have had the strong situation motivation effect, improving the results with several strong extrinsic motivation impulses.

The curriculum created for contact learning does not fully fit distance education. The results here based on a course which even in earlier years was utilising both the concrete hands-on experiments, open learning environments, field education, ICT, and digital and virtual augmented reality solutions [68] gave some encouraging experience for combining formal education and informal, mixed-reality opportunities for modern teacher professional development.

The preliminary results and best practises developed by this study are already benefitted internationally by two major-scale EU-Erasmus+ teacher education projects [69]. Virtual Pathways (www.virtualpathways.eu; accessed 27 January 2022) is unleashing the potential of digital technologies by bridging the gap between formal education and informal learning in science centres and museums in four countries. Discovery Trail (www.avastusrada.ee/en, accessed on 26 December 2022) is utilising the results of this study by facilitating knowledge on how individuals' understandings of complex environmental topics and how knowledge development can best be supported by using digital outdoor learning tools. The project brings together specialists from different disciplines (educational psychology, education, ecology, digital technologies) and several countries that help to widen the scope of information and the experience change. Teachers working together during the project with informal education specialists help to better understand how to support this collaboration best in order to achieve long-lasting changes in students' global challenges related to their understandings.

The results of this study provide new insights for courses that combine lectures and study visits. The aim is not to draw conclusions on one form of teaching over another. In the future, different kinds of combinations may be provided where part of the course can be provided online and part as more traditional contact teaching. The consequences of this study are mainly related to learning and teaching skills. As it is reported above, the teaching students' 21st century skills improved, but some of the results varied. This means that newly graduated teachers are more ready to use digital teaching materials and methods or to teach remotely than ever before, but face-to-face teaching and meeting is also needed. This is an important sign for teacher educators. It gives evidence that hybrid-education should be used, developed, and studied more.

It seems that this research tradition will continue and find new variants still vividly [70] —such as the virus itself—and has changed the educational processes permanently.

**Author Contributions:** Conceptualization, H.S., N.H. and H.T.; methodology, H.S., N.H., L.N. and H.T.; software, N.H., L.N. and H.T.; validation, H.S., N.H., L.N. and H.T.; formal analysis, H.S., N.H., L.N., A.K. and H.T.; investigation, H.S., N.H., L.N., A.K. and H.T.; resources, H.S.; data curation, N.H., L.N. and H.T.; writing—original draft preparation, H.S., N.H. and H.T.; writing—review and editing, H.S., N.H., L.N., A.K. and H.T.; visualization, N.H., L.N. and H.T.; supervision, H.S. and H.T.; project administration, H.S.; funding acquisition, H.S. and A.K. All authors have read and agreed to the published version of the manuscript.

**Funding:** Co-funded by Erasmus+ Programme of the European Union (REF: 2020-1-FI01-KA226-SCH-092545).

**Institutional Review Board Statement:** This article was conducted in accordance with The Declaration of Helsinki, GDPR, and ethical rules & regulations by University of Helsinki.

**Informed Consent Statement:** Informed consent was obtained from all subjects involved in the study.

**Data Availability Statement:** Data partly available by request.

**Acknowledgments:** Open access funding provided by University of Helsinki.

**Conflicts of Interest:** The authors declare no conflict of interest.

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
