# Peer review of "Comparing Contact Education and Digital Distant Pedagogy Strategies: Lockdown Lessons Learnt for University-Level Teacher Education"

_education, doi:10.3390/educsci13020196_

Round 1

Reviewer 1 Report

Paper deals with interesting topic of evaluating effectiveness of university course by comparing effects of different methods used during usual in person (live contact) and online (distant learning) model of teaching because of pandemic. Value of the research lies in a fact that same course content was evaluated in four different periods, before and during pandemic. Results are important for educational theory and practice and can be used in preparation and evaluation od formal and informal courses in educational institutions because there is direct comparison of effects of direct contact and distant learning methods.

In order for paper to be more approachable to teachers and other experts in education there are few suggestions for improvement:

-  Some pedagogical implications and limitations of the research should be mentioned in abstract

-  Section Aim of the present study should be integrated in section research methodology

-  Section Sample should be renamed in research participants because students are not just passive objects

-  Section Findings and section Discussion should be integrated into one section and renamed in Research results and discussion.

- Table 1 and 2 should be more compact and after those tables key highlights of the results should be mentioned with possible discussion and rhetorical questions.

- Figure 2 should be separated into four different figures according to four different periods with proper explanation and highlights.

- General revision of the style of written English and used academic terminology is advised but not necessary

Suggestions are not  mandatory but implementation of them may improve clarity and usability in educational context of this paper    

Author Response

Dear Reviewer 1: On the behalf of our team I really would like to thank You for the fruitful comments that really improved our text – and gave us good food for thought for further discussions !

Answers below  in text with bold font !

Review 1

English language and style

( ) English very difficult to understand/incomprehensible
( ) Extensive editing of English language and style required
(x) Moderate English changes required
( ) English language and style are fine/minor spell check required
( ) I don't feel qualified to judge about the English language and style

Yes

Can be improved

Must be improved

Not applicable

Is the content succinctly described and contextualized with respect to previous and present theoretical background and empirical research (if applicable) on the topic?

(x)

( )

( )

( )

Are all the cited references relevant to the research?

( )

(x)

( )

( )

Are the research design, questions, hypotheses and methods clearly stated?

( )

(x)

( )

( )

Are the arguments and discussion of findings coherent, balanced and compelling?

(x)

( )

( )

( )

For empirical research, are the results clearly presented?

( )

(x)

( )

( )

Is the article adequately referenced?

(x)

( )

( )

( )

Are the conclusions thoroughly supported by the results presented in the article or referenced in secondary literature?

(x)

( )

( )

( )

Comments and Suggestions for Authors

Paper deals with interesting topic of evaluating effectiveness of university course by comparing effects of different methods used during usual in person (live contact) and online (distant learning) model of teaching because of pandemic. Value of the research lies in a fact that same course content was evaluated in four different periods, before and during pandemic. Results are important for educational theory and practice and can be used in preparation and evaluation od formal and informal courses in educational institutions because there is direct comparison of effects of direct contact and distant learning methods.

In order for paper to be more approachable to teachers and other experts in education there are few suggestions for improvement:

-  Some pedagogical implications and limitations of the research should be mentioned in abstract

Ok. Now main teaching opportunities and research challenges added to new version abstract. “As limitation of the study, more students’ prior experience and attitudes with online learning is needed from future research.”

-  Section Aim of the present study should be integrated in section research methodology

These two separate parts now integrated – text became more coherent !

-  Section Sample should be renamed in research participants because students are not just passive objects

Yes: this definition is accurate and now replaced.

-  Section Findings and section Discussion should be integrated into one section and renamed in Research results and discussion.

Yes, good recommendation: It makes the article more coherent and reader friendly. Done now!

- Table 1 and 2 should be more compact and after those tables key highlights of the results should be mentioned with possible discussion and rhetorical questions.

Done. Tables 1&2 are now edited to be more clear – and the results highlighted and discussed in the last Chapter.

- Figure 2 should be separated into four different figures according to four different periods with proper explanation and highlights.

We disagree slightly: While the figures could be more reader-friendly, we think that it is very important that the four time-points can be easily compared. Now in this mode we used -  the comparison is possible!

- General revision of the style of written English and used academic terminology is advised but not necessary

One more native-editor is done.

Suggestions are not  mandatory but implementation of them may improve clarity and usability in educational context of this paper   

Thank you for your feed-back and hints to improve the text !

Dear Reviewer 1: On the behalf of our team I really would like to thank You for the fruitful comments that really improved our text – and gave us good food for thought for further discussions !

Answers below  in text with bold font !

Review 1

English language and style

( ) English very difficult to understand/incomprehensible
( ) Extensive editing of English language and style required
(x) Moderate English changes required
( ) English language and style are fine/minor spell check required
( ) I don't feel qualified to judge about the English language and style

Yes

Can be improved

Must be improved

Not applicable

Is the content succinctly described and contextualized with respect to previous and present theoretical background and empirical research (if applicable) on the topic?

(x)

( )

( )

( )

Are all the cited references relevant to the research?

( )

(x)

( )

( )

Are the research design, questions, hypotheses and methods clearly stated?

( )

(x)

( )

( )

Are the arguments and discussion of findings coherent, balanced and compelling?

(x)

( )

( )

( )

For empirical research, are the results clearly presented?

( )

(x)

( )

( )

Is the article adequately referenced?

(x)

( )

( )

( )

Are the conclusions thoroughly supported by the results presented in the article or referenced in secondary literature?

(x)

( )

( )

( )

Comments and Suggestions for Authors

Paper deals with interesting topic of evaluating effectiveness of university course by comparing effects of different methods used during usual in person (live contact) and online (distant learning) model of teaching because of pandemic. Value of the research lies in a fact that same course content was evaluated in four different periods, before and during pandemic. Results are important for educational theory and practice and can be used in preparation and evaluation od formal and informal courses in educational institutions because there is direct comparison of effects of direct contact and distant learning methods.

In order for paper to be more approachable to teachers and other experts in education there are few suggestions for improvement:

-  Some pedagogical implications and limitations of the research should be mentioned in abstract

Ok. Now main teaching opportunities and research challenges added to new version abstract. “As limitation of the study, more students’ prior experience and attitudes with online learning is needed from future research.”

-  Section Aim of the present study should be integrated in section research methodology

These two separate parts now integrated – text became more coherent !

-  Section Sample should be renamed in research participants because students are not just passive objects

Yes: this definition is accurate and now replaced.

-  Section Findings and section Discussion should be integrated into one section and renamed in Research results and discussion.

Yes, good recommendation: It makes the article more coherent and reader friendly. Done now!

- Table 1 and 2 should be more compact and after those tables key highlights of the results should be mentioned with possible discussion and rhetorical questions.

Done. Tables 1&2 are now edited to be more clear – and the results highlighted and discussed in the last Chapter.

- Figure 2 should be separated into four different figures according to four different periods with proper explanation and highlights.

We disagree slightly: While the figures could be more reader-friendly, we think that it is very important that the four time-points can be easily compared. Now in this mode we used -  the comparison is possible!

- General revision of the style of written English and used academic terminology is advised but not necessary

One more native-editor is done.

Suggestions are not  mandatory but implementation of them may improve clarity and usability in educational context of this paper   

Thank you for your feed-back and hints to improve the text !

Reviewer 2 Report

The novelty and the contribution of this paper is “there”, but some changes are needed. The research approach is weak, and the findings need to be strengthened. Thus, my decision is major revision. Here are my comments on improving the manuscript:

1. Overall:

a)       It seems that this is a longitudinal paper. Why do the authors conduct this study? The research contributions are weak. Please kindly explain.

b)      The research structure is not appropriate for a scientific article because it has only 3 parts including an introduction, results and conclusion. Please update.

c)       Please consider how to effectively integrate some review papers and update.

2. Introduction:

a)       Research questions, that drive the paper, should be built in the introduction from an ongoing and pertinent bibliography (up to 2022-23) and these should be of global interest and not focused on a particular local problem. Identifying a research gap is the most important by indicating in-text some newer references that are significant to your particular field of research.

b)      Due to a weak contribution, please pay attention to addressing new research gaps and then emphasize why the authors do this study.

c)       Please try to provide a more concise Introduction.

3. Discussion:

a)       The proposed reason for conducting such a study is weak due to a lack of explanation and no further explanation regarding how such a work contributes instead of previous ones.

b)      The result analysis is poor and subjective due to a lack of contributions and thorough discussion. Please rewrite it and consider providing a new discussion section to provide significant criticism and research limitations.

c)       Authors should answer your research question in the conclusions and discussion. Please provide a reasonable need to read your work’s results than previous ones or simply answer what we learned compared with current, significant research (up to 2022 should be your work’s “significance”).

d)      How general are your results and how do you believe that such findings have to be of global interest? Please relate these with your limitations and Discussion that is not exist. Why?

e)      Are there any points of view related to the consequences of this study’s limitations that may have an impact on their findings?

f)        Are there any points of view related to the consequences of this study’s limitations that may have an impact on their findings?

g)       Practical and educational implications are not provided.

Author Response

Dear Reviewer 2: Thank you very much for the fruitful comments – which certainly improved the paper and will also give options for further discussions !

See out comments & correction in Italic font below !

English language and style

( ) English very difficult to understand/incomprehensible
( ) Extensive editing of English language and style required
( ) Moderate English changes required
(x) English language and style are fine/minor spell check required
( ) I don't feel qualified to judge about the English language and style

Yes

Can be improved

Must be improved

Not applicable

Is the content succinctly described and contextualized with respect to previous and present theoretical background and empirical research (if applicable) on the topic?

( )

( )

(x)

( )

Are all the cited references relevant to the research?

( )

( )

(x)

( )

Are the research design, questions, hypotheses and methods clearly stated?

( )

( )

(x)

( )

Are the arguments and discussion of findings coherent, balanced and compelling?

( )

( )

(x)

( )

For empirical research, are the results clearly presented?

( )

( )

(x)

( )

Is the article adequately referenced?

( )

( )

(x)

( )

Are the conclusions thoroughly supported by the results presented in the article or referenced in secondary literature?

( )

( )

(x)

( )

Comments and Suggestions for Authors

The novelty and the contribution of this paper is “there”, but some changes are needed. The research approach is weak, and the findings need to be strengthened. Thus, my decision is major revision. Here are my comments on improving the manuscript:

  1. Overall:
  2. a) It seems that this is a longitudinal paper. Why do the authors conduct this study? The research contributions are weak. Please kindly explain.

The longitudinal approach was chosen because we had a unique data to find out what happened before, during, and after the COVID-19 period 2020-21. There is a lot of experiences and result from the pandemic time, but it has not been compared to the situation before and after. Our data was covering also the April years 2019 and 2022 giving evidence based facts about the process and not only describing the pandemic experiences.

  1. b) The research structure is not appropriate for a scientific article because it has only 3 parts including an introduction, results and conclusion. Please update.

The report has seven different parts according the best practices of this Journal. It can be easily seen just by checking the bold subtitles of the article. In addition, the other Revier1 was suggesting and recommendings that the writers will combine chapters “”Aim of the present study should be integrated in section research methodology”” and also two last parts as one with the new name Research results and Discussion, which fits best for scientific reporting.

  1. c) Please consider how to effectively integrate some review papers and update.

Yes. We have done it. See next comment  “2a”.

  1. Introduction:
  2. a) Research questions, that drive the paper, should be built in the introduction from an ongoing and pertinent bibliography (up to 2022-23) and these should be of global interest and not focused on a particular local problem. Identifying a research gap is the most important by indicating in-text some newer references that are significant to your particular field of research.

As the matter of fact, seven first references are from the years 2020-2022. Many of those are even meta-studies based on tens of other basic reports. These reports give also as literature reviews European (by EU) and global (by UNESCO) vision of the main trends. Also many of our references where related to the recent publication of ES-Journal itself. In addition, there is no reason to underestimate earlier reported results: on-line and digital education was not discovered by covid, but there was a lot of  best practices and “lessons learnt” from earlier years, which unfortunately educators were not aware of. We have also wanted to stress the historical perspective of the topic as the online instruction as such is not a new development produced by a pandemic. 2023?! Of course we couldn’t include any references from 2023 – while our submitting happened in December 2022. We have now added still one – very latest report as Jimoyiannis & Koukis, 2023 – to the very end of the Conlusions.

  1. b) Due to a weak contribution, please pay attention to addressing new research gaps and then emphasize why the authors do this study.

We noticed the “research gap” related to COVID-period: We found lots of fine reports from panic reactions immediately and during pandemic – but practically none comparing it also the situation afterward. Our unique data gave opportunity to fill in that gap.

  1. c) Please try to provide a more concise Introduction.

Yes! Thank you for the suggestion. We have edited now texts to “Introduction” based on the points 1abc; and 2abc.

  1. Discussion:
  2. a) The proposed reason for conducting such a study is weak due to a lack of explanation and no further explanation regarding how such a work contributes instead of previous ones.

Ok. We have already commented this in the points 1a; 2a; and also next 3b deals mainly the same topic. This work contributes future actions by creating evidence based best practices bridging the gap between on-line and on-site education.

  1. b) The result analysis is poor and subjective due to a lack of contributions and thorough discussion. Please rewrite it and consider providing a new discussion section to provide significant criticism and research limitations.

Fine. We have now added these elements by rewriting and providing aspects of research limitations by critical practical examples as presented in point “3g”.

  1. c) Authors should answer your research question in the conclusions and discussion. Please provide a reasonable need to read your work’s results than previous ones or simply answer what we learned compared with current, significant research (up to 2022 should be your work’s “significance”).

“Lessons learnt” from our article are clearly differing from many earlier papers, which are not presenting the results after Covid-period. “Recovering” is one of the key elements: everything did not became the same as in the past. New solution were – and were forced – to find. This type of hybrid elements  has to be now utilized not only in schools, but also in real working life and whole society activities. Our study give clear hints for that process. It is also showing that some students do benefit more of this – and some less. To solve this challenge – new teaching methods and combinations are needed. This is also an urgent topic for further research, which we are now encouraged to start.

  1. d) How general are your results and how do you believe that such findings have to be of global interest? Please relate these with your limitations and Discussion that is not exist. Why?

There has not been more global challenge for educational system and research than the pandemic. Thus, systematic findings related to teacher professional practices and routines certainly have more than sporadic value and the globally utilized digital tools are very much the same around the world. Certain elements are common both western countries and other parts of the world.

  1. e) Are there any points of view related to the consequences of this study’s limitations that may have an impact on their findings?

Important point of view! We will certainly add into the text – and also to the abstract – as the other Reviewer suggested. There are always certain limitations related to research strategy and approach chosen. We strongly believe that the main results and trends are reliable and valid in this sense: no test-group vs. control group approach is needed to generalize these results.

  1. f) Are there any points of view related to the consequences of this study’s limitations that may have an impact on their findings?

Exactly the same question as earlier: see 3e.

  1. g) Practical and educational implications are not provided.

We have now added best practices, recommendations and implications for future educational work and also potential further reseach topics (see also 3b, 3c). The consequences of this study are mainly related to learning and teaching skills. As it was said before teacher students 21th century skills improved, but some of the results varied. It means that a newly graduated teachers are more ready to use digital teaching materials and methods or to teach remotely than ever before, but also face-to-face teaching and meeting is needed. This is an important sign for teacher educators. It gives an evidence that hybrid-education should be used, developed and studied more.

Round 2

Reviewer 2 Report

The authors have satisfactorily responded to all my questions and made the necessary changes to the manuscript.